# Factors Associated with Antimicrobial Stewardship Practices on California Dairies: One Year Post Senate Bill 27

**DOI:** 10.3390/antibiotics11020165

**Published:** 2022-01-27

**Authors:** Essam M. Abdelfattah, Pius S. Ekong, Emmanuel Okello, Deniece R. Williams, Betsy M. Karle, Terry W. Lehenbauer, Sharif S. Aly

**Affiliations:** 1Veterinary Medicine Teaching and Research Center, School of Veterinary Medicine, University of California, Davis, Tulare, CA 93274, USA; eabdelfattah@ucdavis.edu (E.M.A.); pekong@ucdavis.edu (P.S.E.); eokello@ucdavis.edu (E.O.); dvmwilliams@ucdavis.edu (D.R.W.); tlehenbauer@ucdavis.edu (T.W.L.); 2Department of Animal Hygiene and Veterinary Management, Faculty of Veterinary Medicine, Benha University, Moshtohor 13736, Egypt; 3Department of Population Health and Reproduction, School of Veterinary Medicine, University of California, Davis, CA 95616, USA; 4Cooperative Extension, Division of Agriculture and Natural Resources, University of California, Orland, CA 95963, USA; bmkarle@ucanr.edu

**Keywords:** antimicrobial drug use, antimicrobial stewardship, dairy cattle, knowledge, logistic regression models, machine learning, survey

## Abstract

Background: The current study is aimed at identifying the factors associated with antimicrobial drug (AMD) use and stewardship practices on conventional California (CA) dairies a year after CA Senate Bill 27. Methods: Responses from 113 out of 1282 dairies mailed a questionnaire in 2019 were analyzed to estimate the associations between management practices and six outcomes including producer familiarity with medically important antimicrobial drugs (MIADs), restricted use of MIADs previously available over the counter (OTC), use of alternatives to AMD, changes in on-farm management practices, changes in AMD costs, and animal health status in dairies. Results: Producers who reported having a veterinarian–client–patient relationship (VCPR) and tracking AMD withdrawal intervals had greater odds of being familiar with the MIADs. Producers who began or increased the use of preventive alternatives to AMD in 2019 had higher odds (OR = 3.23, *p* = 0.04) of decreased use of MIADs previously available OTC compared to those who did not. Changes in management practices to prevent disease outbreak and the use of diagnostics to guide treatment were associated with producer-reported improved animal health. In addition, our study identified record keeping (associated with familiarity with MIADs), use of alternatives to AMD (associated with management changes to prevent diseases and decreased AMD costs), and use of diagnostics in treatment decisions (associated with reported better animal health) as factors associated with AMD stewardship. Conclusions: Our survey findings can be incorporated in outreach education materials to promote antimicrobial stewardship practices in dairies.

## 1. Introduction

Antimicrobial drugs (AMD) are important compounds used in humans and livestock for the prevention, control, and treatment of bacterial infections. Medically important antimicrobial drugs (MIADs) are AMD considered as essential or otherwise important for therapeutic purposes in humans. In the United States, MIADs are AMDs that are important for treating human disease and includes all critically important, highly important, and important drugs according to the Federal Food and Drug Administration’s Guidance for Industry #152 [1]. According to the U.S. Food and Drug Administration [1], for the 2019 calendar year, the cattle industry was involved with 41% of the total sales and distribution of MIADs approved for use in food-producing animals. Likewise, the FDA estimated for the 2019 sales and distribution data that 81% of cephalosporins, 65% of sulfas, 45% of aminoglycosides, and 42% of tetracyclines were associated with cattle. However, AMD use comes with the risk of antimicrobial resistance (AMR) in humans and livestock [2,3,4,5].

In January 2017, the FDA fully implemented the Veterinary Feed Directive (VFD) final rule requiring approval and oversight by a licensed veterinarian for MIADs administered for therapeutic purposes via animal feed to food-producing animals and eliminated non-therapeutic uses for both growth promotion and feed efficiency [6]. The FDA also concurrently applied the same requirement for MIADs administered in drinking water for food-producing animals. Beyond these national regulations, in January 2018, California (CA) implemented Senate Bill 27 (SB 27; codified as the Food and Agricultural Code, FAC 14400-14408, here onwards referred to as SB 27), becoming the first state to require a veterinary prescription, under a valid veterinarian–client–patient relationship (VCPR), for all other dosage forms (e.g., injectables and boluses) of MIADs used for livestock [7]. The SB 27 removed all MIADs from over the counter (OTC) sales for livestock. In addition, SB 27 mandated the development of antimicrobial stewardship (AMS) guidelines and resources that support the collection of information on livestock management practices, the monitoring of AMD sales, and AMD use to provide relevant information to producers and other stakeholders. Similar legislation was passed in other states to increase AMS through the judicious use of AMD, such as Maryland SB 471 [8] and Oregon SB 920 [9]. The implementation of effective AMS practices is critical to reducing the threat AMR poses to animal and public health [10]. Similarly, worldwide AMD use in food-producing animals is increasingly being regulated with the goal of adopting AMS practices. Such regulations have included the ban on the preventive use of AMD in the feed of food-producing animals in the European Union [11], benchmarking AMD use between farms in the Netherlands [12], documenting AMD use on individual farms through electronic connection with billing in Denmark [13] and setting up national targets for a reduction of AMD use in Belgium [14].

Ekong et al. [15] conducted a survey immediately after the implementation of SB 27 to identify the factors associated with the adoption of AMS practices in CA conventional dairies. Amongst the survey findings were that dairy producers who reported keeping written or computerized animal health protocols, keeping a drug inventory log, being aware that the use of MIADs required a prescription, involving a veterinarian in the AMD treatment duration and determination, and using on-farm diagnostics to guide AMD therapy were associated with good AMS practices. Our hypothesis was that factors that were associated with AMS immediately after the implementation of SB 27 were still important a year later. Our objective was to identify the factors associated with AMS practices in adult cows in conventional dairies one year post implementation of SB 27 in CA and to compare them to those identified immediately after SB 27.

## 2. Results

### 2.1. Descriptive Statistics

A total of 131 (10.2%) out of 1282 CA licensed grade A conventional and organic dairies responded to our survey. For the current study, AMS was investigated using responses from only conventional dairies, specifically a total of 113 survey responses. Herd demographics of the dairies included in the survey analyses are summarized in Appendix A. The majority of dairies were from Northern San Joaquin Valley (NSJV; 46.9%) and Greater Southern California (GSCA; 46.0%) compared to Northern California (NCA; 7.1%). The distribution of breeds in respondent dairies was primarily Holstein (56.3%), mixed herds of Holstein and Jersey breeds (35.7%), crossbreed (5.4%), and Jersey (2.4%). 

### 2.2. Logistic Regression Models

For each of the six logistic regression models, the NCA and NSJV regions were combined due to limited sample size and collectively compared to the GSCA region. With a few exceptions, region, herd size, and breed were not significantly associated with the model outcomes.

#### 2.2.1. Predictors Concerning Familiarity of Dairies with the FDA “MIADs” Term

Among the dairies that completed the survey, a total of 73 respondents (65.8%) reported familiarity with FDA’s term ‘MIADs’, while 38 respondents (34.2%) were not familiar with this term. Table 1 summarizes the final logistic regression model for the association between the survey factors and familiarity with MIADs. Dairy producers who reported tracking both milk and meat withdrawal intervals, amongst other information such as date, dose, or route of AMD treatment, had greater odds (OR = 10.6, *p* < 0.01) of being familiar with MIADs compared to those who did not track either. Among the dairies in our study, 94 dairies (84%) selected milk and/or meat withdrawal intervals as information they tracked during AMD treatment and 18 dairies (16.1%) reported tracking either date, dose, or route of AMD treatment. Responses to the related question of “Do you keep track of AMD withdrawal intervals for treated cows?” confirmed a similarly high response rate with 99% of respondents reporting that they keep track of AMD withdrawal intervals, with more than half (58%) using computer software while the remaining (41%) use either paper records, white/chalk board, memory, or markings on animals. In addition, dairy producers who reported having a VCPR had greater odds (OR = 15.3, *p* = 0.03) of being familiar with MIADs than those who reported they did not have a VCPR. 

#### 2.2.2. Predictors Concerning the Use of MIADs That Were Restricted from OTC Sales Beginning in January 2018

In our survey, 81 of the 113 respondent dairies reported use of OTC MIADs before 2018. However, of the 81 respondents, only 78 dairies responded to a survey question regarding the changes (decreased, no change, or increased) they made for use of OTC MIADs after SB 27 became effective. Of those 78 dairies, 37 (47.4%) reported decreased use of MIADs federally labeled for OTC sale, while 41 dairies (52.6%) reported increased or no change of this category of MIADs. The final model for the associations between the survey factors and the change in use of MIADs that were previously available OTC is summarized in Appendix A. Large dairies (≥1304 cows) had six-times greater odds for the decreased use of OTC MIADs compared to small dairies (<1304 cows; *p* < 0.01). In addition, our model showed that producers who began using or increased the use of preventive alternatives to AMD in their dairies in 2019 had higher odds (OR = 3.2, *p* = 0.04) of a decreased use of MIADs that were previously available OTC compared to those who did not. For producers who reported the use of preventive alternatives to AMD in our study, 58.3% reported the use of vaccines, herbal remedies (36.1%), vitamins (66.7%), and/or minerals (38.9%).

An analysis of the survey responses from producers who began or increased their use of preventive alternatives to AMD in 2019 showed that a higher percentage of such producers made changes in management to prevent disease outbreaks or spread compared to those who neither began nor increased their use of such alternatives (44.8% ± 9.4 vs. 23.8% ± 4. 8; *p* = 0.03). Furthermore, a higher percentage of such producers vaccinated their animals against coliform mastitis compared to those who neither began nor increased their use of AMD alternatives (92.6% ± 5.13 vs. 74.3% ± 5.11; *p* = 0.04).

#### 2.2.3. Predictors Concerning the Use of Preventive Alternatives to AMD on Dairy Farms

Among the dairies who completed the survey, a total of 30 respondents (26.8%) reported an increased use of preventive alternatives to AMD in their dairies during 2019. Table 2 summarizes the final model for the survey factors associated with the use or increased use of preventive alternatives to AMD.

Our model showed that producers who received training or participated in any dairy quality assurance programs during 2019 had greater odds (OR = 3.6, *p* = 0.05) of using or increasing the use of AMD preventive alternatives compared to those who did not participate in such programs. In addition, our survey revealed that conventional dairies that reported a decreased use of MIADs previously available OTC had higher odds (OR = 6.0, *p* = 0.01) of usage or an increase in the use of AMD preventive alternatives compared to dairies that reported increased or no change in the use of MIADs previously available OTC. 

#### 2.2.4. Predictors Concerning Changes in Management Practices to Prevent Spread or Outbreaks of Disease on Dairies

Among the dairies who completed the survey, a total of 32 respondents (29.4% ± 4.4) reported they had made changes in management to prevent disease outbreaks or spread in their dairies, while 77 respondents (70.6% ± 4.4) reported they had made no changes in management. In our study, 22% of respondents reported a change in management in the form of vaccination programs to prevent disease, 4.6% reported a change in management in the form of improved biosecurity (e.g., restricted traffic on operation, better isolation of sick animals, or designated separate equipment for feed and manure handling), 1.8% reported a change in management in the form of prepurchase testing of animals before being added to the herd, and 0.9% reported a change in management in the form of quarantining of purchased/returning animals from offsite locations.

Appendix A summarizes the final model for the survey factors associated with changes in management to prevent outbreaks. Our model showed that producers who included veterinarians in the revision of animal health protocols for cows had greater odds (OR = 13.5, *p* = 0.01) of reporting changes in management practices in their dairies compared to those who did not include veterinarians in protocol reviews. In addition, producers who reported better animal health on their farms during 2019 had greater odds (OR = 4.7, *p* < 0.01) of having made management changes to prevent disease spread or outbreaks compared to those who reported no changes to their animal health. Finally, producers who reported the use or increased use of alternatives to AMD in their dairies, such as vitamins, minerals, and vaccines, to reduce the use of MIADs were at greater odds (OR = 2.7, *p* = 0.04) of having made management changes to prevent disease spread or outbreaks compared to those who did not use any alternatives to AMD.

#### 2.2.5. Predictors Concerning Change in a Dairy’s AMD Costs

Changes in a dairy’s drug costs were dichotomized as “decreased AMD costs” vs. “increased/no change” in AMD costs. Among the dairies who completed the survey, a total of 29 respondents (26.6% ± 4.3) reported decreased farm AMD costs since January 2018, while 80 respondents (73.4% ± 4.3) reported increased or no change in farm AMD costs. Appendix A summarizes the final model for the survey factors associated with decreased farm AMD costs. 

Our model showed that producers who reported the use or increased use of AMD preventive alternatives reported decreased AMD costs in their dairies compared to those who did not use AMD preventive alternatives (OR = 7.89; *p* ≤ 0.01). In addition, our study showed that producers who indicated better animal health in their farms also reported decreased AMD costs in their dairies compared to those who reported no change or worse animal health (OR = 5.48; *p* ≤ 0.01). 

#### 2.2.6. Predictors Concerning Change in Reported Farm Animal Health

The response to a survey question regarding the farm animal health status as an outcome was dichotomized as “better animal health” vs. “worse/no change”. Among the dairies who completed the survey, a total of 47 respondents (44.3% ± 4.9) reported better animal health since January 2018, while 59 respondents (55.7% ± 4.9) reported worse or no change on the farm’s animal health. Table 3 summarizes the final model for the survey factors associated with better animal health in the dairy farms.

Our results showed that producers who included the veterinarian in the decision as to which AMD were used to treat sick cows had lower odds of reporting better animal health in their dairies compared to those who did not include the veterinarian in their treatment decision (OR = 0.31; *p* = 0.01).

Producers who indicated using on-farm diagnostic techniques to guide AMD treatment decisions had greater odds of reporting better animal health on their dairies compared to those who did not use diagnostic techniques (OR = 4.53; *p* < 0.01). 

In addition, the current model indicated that producers who reported management changes in their dairies to prevent disease spread or outbreaks had greater odds (OR = 2.91, *p* < 0.01) of having better animal health in their farms during 2019 compared to those who reported no changes in management practices. Similarly, producers who reported decreased AMD costs in their dairies had greater odds (OR = 3.68, *p* = 0.01) of having better animal health in their farms during 2019 compared to those reported increased/no change in their AMD costs. 

#### 2.2.7. Predicting Factors Associated with Dairy Producers’ Perceptions Regarding the Importance of AMS Practices on Dairies Using Machine Learning Classification Models

The distribution of CA dairy producers (*n* = 113) with respect to five statements on the importance of AMS practices is presented in Figure 1. Based on the number of the AMS practices that the dairy producers scored as very important, we found that 41.3%, 37.6%, 19.3%, 0.9%, and 0.9% of producers were given a score of 5, 4, 3, 2, and 1, respectively. By classifying producers as having “good to excellent = score of 4 and 5” AMS knowledge or as having “limited-moderate = score of 3 or less” AMS knowledge, we found that 78.9% (86/109) of producers had “good to excellent” knowledge, while 21.1% (23/109) of producers were classified as having “limited-moderate” knowledge based on their responses. 

A descriptive analyses of responses based on their knowledge of AMS practices (Appendix A) showed that a greater percentage of respondents classified as having good to excellent AMS knowledge kept drug inventory logs compared with respondents classified as having limited–moderate knowledge (81.4% ± 5.9 vs. 18.6% ± 5.9; *p* < 0.01). Similarly, results showed that a greater percentage of respondents classified as having good to excellent AMS practices knowledge were familiar with FDA’s term “MIADs” compared with respondents classified as having limited–moderate knowledge (90.1% ± 3.5 vs. 9.85% ± 3.5; *p* < 0.01).

Appendix A summarizes the performance of three different classification algorithms. The three models had similar specificity; however, the highest sensitivity was achieved by the gradient boosting (GB) model. The three classification models identified herd size and familiarity with MIADs as the top two important predicting variables for classifying dairy producers’ AMS knowledge. The decision tree (DT) model showed that a higher percentage of dairy producers who were familiar with the FDA term MIAD were classified as producers with “good-excellent” AMS knowledge compared to producers with “limited-moderate” AMS knowledge. In addition, the DT model further classified dairies of a herd size >1737 milking cows/herd as dairies with “good-excellent “AMS practices compared with dairies of a herd size ≤1737 cows/herd. 

Despite the moderate sensitivity (46.5%), random forest had the greatest precision (88.9%) compared to the remaining ML models. In general, the best classification performance as measured using the AUC was obtained using the GB model (AUC = 96). The top ten GB-identified predicting variables included herd size, familiarity with the FDA term “MIADs”, annual RHA for milk production, location of herd in CA, veterinarian input on parenteral AMD purchases, willingness to treat animals with AMD alternatives, use of on-farm diagnostic techniques to guide AMD treatment decisions, producer agreement with the statement that antibiotics use in livestock may cause problems in humans, and the basis for the mastitis treatment decision (abnormal milk, California Mastitis Test, microbiologic culture of milk samples, drug sensitivity testing or treatment while culture results are pending) and producer agreement that current antibiotic use in livestock will make it harder to treat future infections (Appendix A). 

## 3. Discussion

The response rate obtained in this survey was relatively low; however, our response rate is consistent with other mailed surveys conducted in CA [16,17] and with other surveys of both Canadian [18] and UK dairies [19]. Furthermore, response rates stratified by region in our survey were like the regional distribution of herds as reported in a California Department of Food and Agriculture report [20]. The current survey respondents reported a greater proportion of mixed-breed herds compared to a previous survey, indicating either a different population of respondents or a shift in the state dairy herd breed make-up [21]. Earlier surveys also reported higher estimates for the Holstein herd composition (65% in Aly et al. [22]; 77% in Love et al. [16]; and 66% in Ekong et al. [23]) compared to our survey. Such variability may be due to inherent differences in producers responding to such a wide range of surveys (beef quality assurance, bovine respiratory disease, and antimicrobial stewardship), which could be influenced by their interests.

### 3.1. Predictors Concerning Familiarity of Dairies with the FDA “MIADs” Term

Our results showed that dairy producers who reported tracking both milk and meat withdrawal intervals during AMD treatment or those who reported having a VCPR were more familiar with the FDA’s term “MIADs” than those who did not report. These findings agree with findings from the survey conducted the year SB 27 was implemented [15]. However, our survey showed that 34.2% of the dairy respondents were not familiar with or not sure how the FDA’s term “MIADs” related to their dairies. Similarly, a survey that was conducted to identify the common perceptions of Tennessee cattle producers regarding the VFD [24] found that 13% and 25% of dairy producers were not familiar at all or were slightly familiar with VFD, respectively. Hence, more educational outreach is needed to increase producers’ knowledge and familiarity with the judicious use of MIADs in maintaining the health of cattle and reducing the pressure for AMR. The veterinarians can play an important role in communicating and educating farmers on concepts of AMR and the judicious use of AMD at the farm level. The survey conducted in the UK [19] found that the dairy farmers who had greater awareness of AMR were those that had more visits from and contact with their vets.

The majority of respondents (94.6%) in our survey confirmed they had a VCPR, while the remaining respondents (5.4%) indicated they did not; however, the latter also indicated that a veterinarian was involved in AMD treatment decisions, which may refer to the need for outreach on what establishes a valid VCPR. Veterinarians can play a major role in educating dairy producers to ensure that they have knowledge of AMR threat and AMS. Alternatively, producers may gain knowledge of MIADs independent of their herd veterinarian due to their own education and inquisition. Nevertheless, under the recently adopted VFD final rule and SB 27 regulations, veterinarians are expected to work alongside livestock producers as well as feed manufacturers and distributors to assume a greater role and increased responsibility for the use of MIADs. The VFD final rule specifies that the veterinarian must work with their client and assume responsibility for making clinical judgments about animal health; must have sufficient knowledge of the animals by virtue of examinations and/or visits to the farm where the animals are located; and must provide for any necessary follow-up evaluation or care, which are necessary components of a VCPR [6].

### 3.2. Predictors Concerning the Use of MIADs That Were Restricted from OTC Sales Beginning in January 2018

On January 2018, SB 27 moved all MIADs that were federally labeled for OTC sale, such as penicillin, oxytetracycline, and tylosin, to prescription status in CA. Our model showed that herd size and the use of preventive AMD alternatives were important predictors for dairy producers reporting a decreased use of OTC AMD in their dairies. The reported decrease in the use of OTC MIADs by large dairies compared to smaller dairies could be attributed to the differences in management practices, presence of computerized health protocols, and familiarity with MIADs or the use of preventive alternatives to MIADs. Our machine learning models also showed that the majority (84%) of the large dairy producers were classified as producers with “good to excellent” AMS knowledge compared with producers of small dairies. Similarly, the survey by Ekong et al. [15] found that large dairies producers were more familiar with MIADs, reported a greater use of alternatives to AMD, and reported better animal health compared to small dairy producers after implementation of SB 27.

The use of preventive alternatives to AMD may have filled the gap in the decreased use of OTC MIADs; however, it is not known whether the increased use of alternatives preceded or followed the decrease in the use of AMD previously available OTC. Similarly, Ekong et al. [15] found that producers of large dairies and producers who began using AMD alternatives had higher odds for reporting a decreased use of MIADs that were previously available OTC compared to producers of small dairies or producers who did not use alternatives to AMD.

In addition, most producers (53%) who reported a decreased use of OTC MIADs also reported that they strongly agreed that current AMD use practices in animal agriculture will make it harder to treat future livestock infections compared to producers who did not report a decreased use of AMD. Such findings indicate that producers who believe that current AMD use practices will make it harder to treat future livestock infections may have also employed good management strategies that include disease prevention and outbreak investigations, and the use of preventive alternatives, which may result in decreased AMD use on dairy farms. A study that surveyed the New York State dairy veterinarians’ perceptions of AMD use [25] reported that veterinarians in their study believed that AMD use could be reduced through improved herd management strategies. However, the veterinarians also stated that the biggest barrier to implementing these changes were financial. Gerber et al. [26] found that dairy producers who implemented preventive management changes for udder and uterine health significantly reduced the use of systemic AMD for both udder and uterine health while maintaining animal health compared to producers that did not implement improvements in herd health management.

### 3.3. Predictors Concerning the Use of Preventive Alternatives to AMD on Dairy Farms

Our results showed that dairy producers who reported a decreased use of OTC AMD and reported participation in dairy quality assurance programs were more likely to report the use or increased use of preventive alternatives to AMD in their dairies. Dairy quality assurance programs are voluntary programs that promote quality animal care practices, food safety and quality assurance, as well as enhanced consumer confidence in dairy products (CDRF, 2011). These programs, including CDQAP (California Dairy Quality Assurance Program), the National Dairy FARM (Farmers Assuring Responsible Management) program, cooperative extension outreach education, creamery-led programs, and on-farm training, provide training and standards for quality animal care to promote best management practices for AMS and public health. Most of these programs have specific modules about AMS which provide ongoing education for the dairy community on the judicious and responsible use of MIADs, including avoidance of drug residues in milk and meat.

Similar results were obtained by Ekong et al. [15] who found that producers who reported a decreased use of OTC MIADs had 5.2-times greater odds of having initiated the use or increased use of alternatives to AMD in their dairies compared to producers who made no change or increased use of OTC MIADs. Antimicrobial drug preventive alternatives may have replaced MIADs that were previously available OTC, as evident by the reported decrease in the use of AMD for therapeutic purposes by producers who used or increased the use of alternatives. Alternatively, producers could have increased their use of AMD alternatives due to other reasons including changes in costs of MIADs that may have favored a shift in use of such alternatives. Therefore, further research is needed to verify our findings and determine the success of such preventive measures as alternatives to AMD.

### 3.4. Predictors Concerning Changes in Management Practices to Prevent the Spread or Outbreak of Disease in Dairies

Our results showed that dairy producers who reported the inclusion of veterinarians in the revision of health protocols were more likely to report changes in management practices. Herd veterinarians are professionals with experience in animal health and direct knowledge of their herds, and hence are capable of identifying the successful interventions to control and prevent diseases. The survey conducted in New York exploring dairy farmers‘ attitudes [27] indicated that dairy farmers are more receptive to the opinions of fellow farmers and veterinarians regarding AMD use compared to scientists/researchers and government regulators. A valid VCPR between the dairy producer and herd veterinarian can improve the health and welfare of animals by identifying shared concerns and adopting action plans [28]. For example, Jansen et al. [29] found that the communication between farmers and veterinarians helped in the adoption of practices that reduced mastitis.

Our survey findings also showed that changes in management practices were associated with better animal health. Implementing a disease prevention plan that includes good hygiene, isolation of sick or new animals, and a regular vaccination program are associated with improved animal health and productivity [30]. Good management practices help in reducing disease incidence through inhibiting the proliferation of, exposure, or susceptibility to pathogens [31]. Good management practices for biosecurity focus on efforts to prevent the entry of diseases onto the farm (external biosecurity) as well as to prevent disease transmission within the farm (biocontainment) [32,33]. Ohlson et al. [34] found an association between the lower prevalence of BRD infections with better biosecurity at the herd level. There is strong evidence that high standards of both biocontainment and external biosecurity may lead to improved animal health and, in turn, to a reduction of AMD use [30,35].

Producers who reported making changes in management to prevent disease outbreaks or spread in their dairies also reported an increased usage of alternative preventive measures to reduce the use of AMD. Preventive alternatives may work by improving animal health and hence reducing the need for AMD. However, more research is needed to identify alternatives with the same effectiveness and safety for dairy cattle in comparison to AMD [36].

### 3.5. Predictors Concerning Change in Dairy’s AMD Costs

Producers reported that better animal health and use or increased use of AMD alternatives were important predictors of decreased AMD costs in their dairies. Increased use of preventive alternatives to AMD in livestock production has been associated with improved animal health and reduction in both AMD use and AMR [31,37]. The use of vaccines and other preventive measures can help minimize the need for AMD by preventing and controlling infectious diseases in animal populations [31]. Several studies have demonstrated that the use of various bacterial as well as viral vaccines in animals can result in a significant reduction in AMD consumption [31,38,39]. Furthermore, improved animal health may be associated with decreased AMD use and consequently reduced AMD cost. Maximizing the management practices that promote animal health and reduce the incidence of diseases may decrease the use of MIADs in dairy cattle, which is an important factor in reducing the pressure for AMR [40]. In agreement with our results, Ekong et al. [15] found that the reported use or increased use of AMD preventive alternatives was a predictor for a reported decrease in farm AMD cost, their study also reported that inclusion of veterinarian in decision to use AMD, decreasing use of MIADs previously available OTC, participation in dairy quality assurance programs were positively associated with reporting decreased farm AMD cost.

### 3.6. Predictors Concerning Farm’s Animal Health Compared to 2018

Our findings showed that the use of on-farm diagnostic techniques reduced AMD cost and adopting changes in management to prevent disease outbreak/spread were positively associated with better animal health. In agreement with the findings reported by Ekong et al. [15], our survey producers who reported decreased AMD costs on their dairies had greater odds (OR = 4.57, *p* = 0.01) of having better animal health in their farms during 2019 compared to those who reported increased/no change in their AMD costs. Both surveys also identified the use of on-farm diagnostic techniques to guide AMD treatment and improved management practices to prevent both disease outbreak and spread as predictors of producers who reported better animal health. The use of diagnostic techniques such as laboratory culture, auscultation, and lung ultrasound to guide treatment decisions for cows is an important practice to facilitate the judicious use of AMD [41,42]. The availability of affordable diagnostic tools that can detect animals at early stages of disease is important to prevent the high financial costs derived from lost productivity and the treatment of diseased animals [43]. Testing animals for disease can boost herd health and cut costs associated with AMD treatment [44]. A UK survey of dairy herds reported that both farmers and veterinarians recognized there was substantial room for improvement of current diagnostic tools for the detection of mastitis and metabolic disease in dairy cows, particularly with regard to early disease detection [44].

### 3.7. Comparing Survey Findings Immediately Post SP 27 (2018) and One Year Later (2019)

A detailed comparison of the associations explored over the two AMS surveys conducted in CA post SB 27 shows specific shared predictors with similar magnitudes of associations (Appendix A). Predictors common between the current survey (2019) and a year prior (2018) include the reported decreased use of AMD, increased use of AMD alternatives, having a VCPR, tracking AMD treatment information, on-farm record keeping, the use of on-farm diagnostic techniques to guide AMD treatment, participation in dairy quality assurance programs, the inclusion of veterinarians in decisions for the selection of AMD to treat sick cows, and implementation of management practices to prevent disease introduction and spread. In agreement with the current survey (2019), surveys conducted on Illinois dairies [45] and Ohio dairies [46] identified appropriate antimicrobial treatment selection; the use of health protocols; on-farm record keeping; and the application of herd-specific veterinary written protocols, education, and training of farm personnel on diagnostic criteria for the initiation of AMD treatments as important areas for the improvement of AMS in dairies.

### 3.8. Factors Associated with Dairy Producers’ Perceptions Regarding the Importance of AMS Practices on Dairies

Our main goal for machine learning (ML) classification models was to identify factors that can assign dairy producers with good–excellent AMS knowledge. Machine learning models have been used in human medicine to predict AMS intervention in hospitals [47]; however, their use lacks prediction of AMS in veterinary medicine. Using such tools can be helpful for the prediction of AMS interventions in the dairy industry. Machine learning models can be developed to predict which dairies require a stewardship intervention. However, further work is required to develop models with adequate discriminatory power to be applicable to the real-world dairy industry. Our ML models classified producers of large dairies (>1737 milking cows/herd) as producers with “good-excellent” AMS knowledge compared with producers of small dairies (≤1737 cows/herd). A recent Australian study [48] surveyed livestock veterinary practices and identified a lack of access to education, training, and AMS resources as key barriers to the implementation of AMS practices in veterinary practices. Jones et al. [49] suggested that the scientific advice to convey to dairy farmers to achieve responsible AMD use would include advice on best farm practice to minimize the risk of disease and data on cost savings that might be obtained from reduced AMD. In agreement with Ekong et al. [15], our classification ML models also showed that the knowledge of dairy producers about AMS and the adoption of AMS programs may be improved using continuing education and outreach specifically to producers of small and medium-sized dairies. Our results identified that herd size, familiarity with MIADs, RHA, the location of the dairy in CA, tracking AMD treatment, having a written computerized protocol, the somatic cell count, vaccination of animals against mastitis, keeping a drug inventory log, and awareness that MIADs require prescription were the common predictors for classifying dairy producer knowledge regarding AMS practices.

The majority of conventional dairy producers in our study (>95%) indicated that the administration of the appropriate AMD dose, the route, good record keeping on AMD treatment, and observing AMD withdrawal periods are very important AMS practices. This agrees with a previous study that has suggested that improvement in dairy farmers’ AMD use practices can be achieved by using written treatment protocols [50] and the appropriate use of AMD through antimicrobial testing to guide the AMD treatment [51]. Therefore, more extension and outreach efforts should focus on those components to improve antimicrobial stewardship in California dairies.

### 3.9. Study Limitations

The response rate of the current survey was relatively low; however, it is similar to other surveys conducted in California. Furthermore, it is possible that only the producers most interested in AMS responded to this survey, therefore our survey could be subject to different selection bias. As with any survey, our findings are limited by the responses obtained through a questionnaire mailed to CA conventional dairy producers, which may be subject to information bias. However, we piloted the questionnaire using in-person interviews with extension and outreach specialists and veterinarians in CA [21] and used both multiple-point scales and ordinal Likert scales. The actual AMD usage or treatment practices and the health status of the cows on respondent dairies were not measured and hence findings based on related questions should be interpreted with caution. Finally, AMS practices cannot be characterized by the current survey’s responses alone. Therefore, further studies are needed to directly measure the associations between AMD use and AMS practices based on management protocols and an evaluation of both health and production records.

## 4. Materials and Methods

A survey instrument was developed to collect information on AMD use in adult cows in CA dairies during 2019, one year post full implementation of SB 27. The survey was mailed to 1282 grade A dairies in CA during the period from May to December 2019. A list of all licensed grade A dairies in California in 2017 was obtained from the California Department of Food and Agriculture and served as a sampling frame for our survey. The survey development, descriptive statistics, and cluster analyses of the 2019 survey are described in detail in Abdelfattah et al. [21]. Briefly, the survey questionnaire consisted of 44 questions partitioned into three main sections. The first section included variables about herd demographics including the respondent’s role, the county where the dairy was located, the herd’s breed(s), number of milking cows, annual rolling herd average (RHA) milk production, and previous month’s average bulk tank somatic cell count. The second section included questions about dairy cow health management and AMD use including protocols for dry-off, vaccination, disease prevention and diagnosis, sources of information on AMD, who makes decisions on the AMD purchased and used, whether producers had written or computerized animal health protocols, use of a drug inventory, and the presence of a VCPR in dairies. The third section inquired about the dairy practices including enrollment in the animal welfare audit and/or dairy quality assurance programs, producer’s familiarity with MIADs and changes made since 2018 with regard to MIADs previously available OTC, AMD costs, the use of AMD alternatives, disease prevention, and the herd’s health status. An optional comments section was included in the survey to allow respondents to provide feedback about their concern regarding AMD use and AMR in dairies. Multiple-point scales and ordinal Likert scales were used to capture the participant responses to the survey questions.

### 4.1. Statistical Analyses

#### 4.1.1. Descriptive Statistics

For the 113 conventional survey responses, proportions and their standard errors were computed for categorical and ordinal variables, while means and standard errors were computed for continuous variables. Data on the location of dairies in CA were categorized into three regions, namely Northern California (NCA), Northern San Joaquin Valley (NSJV), and Greater Southern California (GSCA), based on the distinct differences among the three regions in dairy infrastructure and management practices [16]. For the purposes of model building, milking herd size was categorized as ≤1304 or >1304 milking cows based on the CA mean herd size while RHA was categorized as <10,880 kg/cow or >10,880 based on the CA mean herd milk production [20].

#### 4.1.2. Logistic Regression Models

Six survey questions related to AMD use and AMR in CA dairies were selected and analyzed as outcome (dependent) variables. These outcome questions included:(1)Familiarity of dairy producers with the FDA’s MIADs term. The familiarity of dairy producers with MIADs was identified if the survey respondent recognized the FDA classification of MIADs as important, highly important, or critically important drugs, and/or that MIADs are available for livestock only via prescription or veterinary feed directive pursuant to VCPR with a licensed veterinarian. Familiarity with MIADs was dichotomized into 2 levels: “familiar” and “not familiar.”(2)Changes made since January 2018 regarding the use of injectable, bolus, and/or intramammary dosage forms of OTC MIADs. This outcome was classified as “decreased OTC MIADs use” or “increased or no change in the use of OTC MIADs”.(3)Initiation or increased use of alternatives to AMD since January 2018. This third outcome was dichotomized as “yes” (use AMD alternatives) or “no” (do not use AMD alternatives).(4)Changes in management practices to prevent disease outbreak or spread since January 2018. This fourth outcome was dichotomized as “yes” (made changes in management practices in the form of improvement in vaccination programs, quarantined purchased and returned animals from offsite locations, improvements of farm biosecurity measures, or testing of pre-purchased animals for infectious diseases before joining the herd) and “no” (no changes in management practices).(5)Description of the farm’s AMD costs since January 2018. This fifth outcome modeled the changes in the farm AMD drug costs in 2019 and was dichotomized as “decreased AMD cost” or “increased/no change AMD cost”.(6)Description of the farm’s animal health conditions since January 2018. This sixth outcome modeled the changes in farm animal health and was dichotomized as “better animal health” or “worse/no change in animal health”.

Logistic regression models were specified for each outcome. Univariate models were used to assess the association between the survey variables described in Abdelfattah et al. [21] and each of the six outcomes of interest. Predictors associated with an outcome of interest at *p* ≤ 0.20 were considered for further modeling. A manual forward model building approach was used while assessing confounding by the breed, herd size, RHA, and region using the method of change in estimates [52]. Previously excluded variables were offered into the model again and retained at *p* ≤ 0.05. All biologically meaningful interaction terms were explored using significance testing. Collinearity of all the potential explanatory variables was checked using the spearman rank correlation statistic. The diagnostics were performed, and plots of residuals were examined, confirming the goodness of fit of each model. Final model selection and fit were assessed using the Akaike Information Criterion (AIC). The coefficients, odds ratios (OR), and their associated 95% confidence intervals were estimated in the final logistic model for factors statistically significant (*p* ≤ 0.05) with the outcome. All statistical analyses were performed using Stata 15 (Stata Corp, College Station, TX, USA).

#### 4.1.3. Machine Learning Classification Models

Machine learning algorithms were used to predict dairy producers who considered AMS practices as important to preserve the efficacy of AMD and reduce AMR in dairy farms. In this study, three widely used machine learning algorithms were evaluated, including decision tree (DT), random forest (RF), and gradient boosting (GB) algorithms. The responses of 113 conventional dairy producers to survey questions about AMS practices were used as a target for the three predictive models. The question requested respondents to classify the following five AMS practices as very important, somewhat important, or not important: (1) administration of appropriate AMD dose, route, and duration; (2) good record keeping on treatment and treatment dates; (3) having a current VCPR; (4) observing withdrawal periods and drug residue avoidance; and (5) using alternatives to AMD such as vaccines and supplements. The respondents were given a score of one to five based on the number of the AMS practices that they scored as very important. For example, if one respondent indicated that all five previously mentioned AMS practices are very important, they would be given a score of 5 and so on. A score of 5 was ranked “excellent”, 4 as “good”, 3 as “moderate”, and 2 or 1 as “limited” knowledge of AMS practices. Then, dairy respondents were reclassified as having “good to excellent” AMS knowledge based on a score of ≥4 or as having “limited-moderate” knowledge based on a score of ≤3. Having “good to excellent” vs. “limited-moderate” AMS knowledge was the outcome of the three predictive models (target variable). Each model was offered a set of 27 survey factors (predictor variables) that contributed to most of the variability in the survey responses, as identified using multiple factor analysis, as reported by Abdelfattah et al. [21]. The predictor variables included herd demographics (herd size, location, annual rolling herd average milk production, and somatic cell count); good general practices (feed newborn calves colostrum from fresh cows, have a separate calving pen, and vaccinate against different diseases); AMD usage information (sources of information about AMD, inclusion of veterinarian in the decision to purchase and treat cows with AMD, tracking of AMD withdrawal periods, and having a written/computerized health protocol); mastitis management practices (basis for treatment of mastitis with AMD, AMD treatment choice, and class of AMD used to treat mastitis); metritis management practices (basis for treatment of metritis and AMD treatment choice); treatment choice for pneumonia in adult cows (oral or injectable AMD); familiarity of producers with the FDA’s term “MIADs” and that MIADs require prescription; and the level of agreement (strongly agreed/agreed, neutral, or strongly disagree/disagree) of dairy producers on the following statements: current antibiotic use practices in animal agriculture will make it harder to treat future livestock infections, antibiotic use in livestock does not cause problems in humans, antibiotic use in livestock leads to bacterial infections in people that are more difficult to treat, any use of antibiotics may result in infections that are more difficult to treat in the future, and willingness to treat animals with alternatives to antibiotics if they were equally effective and comparable in price. The original dataset was partitioned into training and testing data sets using a random split ratio of 70: 30 (training: test). Each model was trained with the training dataset and evaluated by assessing their predictive performance on the testing dataset using Salford Predictive Modeler 8.0 software (https://www.minitab.com/en-us/products/spm/ accessed on 15 July 2021). For the DT algorithm, we used a 10-fold cross-validation method for testing, Gini as the optimization method, and the minimum cost tree as the choice for the best tree [53]. The RF method is based on multiple decision trees: it builds several individual classification trees using a random subsample of the data and then selects the most popular class [54]. For the RF model, we used the out-of-bag testing method with 500 classification trees. Meanwhile, for the GB model, we used a 10-fold cross-validation method, a tree size of 500, balanced sample weights, and the best model chosen by cross-entropy [54]. Evaluation of model performance was based on accuracy, sensitivity, specificity, F1 score, and the Area Under Curve (AUC) estimated from receiver operator characteristic (ROC) curve analyses [55]. The probability threshold at which a classification was made was initially set at a standard 0.5. Then, the obtained results were explored, and the optimal predictive threshold was determined to select the highest specificity [56].

## 5. Conclusions

The majority of survey producers agreed that the administration of the appropriate AMD dose, the route, keeping good records on AMD treatment, and observing AMD withdrawal periods are very important AMS practices. Our results showed that better cow health was positively associated with management changes to prevent disease spread and the use of on-farm diagnostic techniques to guide AMD treatment decisions. In addition, the use or increased use of preventive alternatives to AMD was positively associated with the decreased use of OTC AMD and decreased farm AMD cost. Our findings identified that factors that were associated with antimicrobial stewardship practices in conventional CA dairies immediately after implementation of SB 27 are still important one year later. These factors included a valid veterinarian–client–patient relationship, a reduction in the use of MIADs that were previously available OTC before implementation of SB 27, a decreased farm AMD cost, and the use or increased use of AMD alternatives. In addition, herd size, familiarity with MIADs, location of the dairy in CA, tracking AMD treatment, having a written computerized protocol, keeping drug inventory logs, and awareness that MIADs require prescription were the common predictors between the survey of 2018 and 2019 that identified dairy producers with “good to excellent” antimicrobial stewardship knowledge using machine learning. The findings of our survey should be interpreted with caution due to biases related to surveys.

Findings from this survey may benefit extension outreach efforts by offering education and training on identified areas associated with improved antimicrobial stewardship practices in CA dairies. Future research is needed to study the association between the implementation of antimicrobial stewardship practices and the reduction in the prevalence of AMR in food-producing animals. In addition, further research is needed to identify the barriers that prevent the implementation of the identified components of antimicrobial stewardship practices in CA dairies.

## Figures and Tables

**Figure 1 antibiotics-11-00165-f001:**
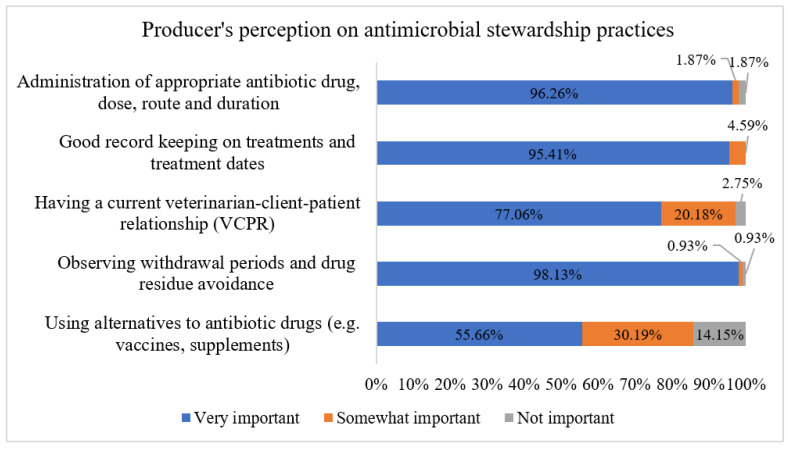
Distribution of responses from Californian conventional dairy producers to five statements on importance of antimicrobial stewardship practices. Bars represent proportion of responses by level of importance. The plot summarizes survey responses from 113 conventional dairy respondents.

**Table 1 antibiotics-11-00165-t001:** Estimated coefficients and odds ratios from a multilevel mixed-effects logistic regression model for the association between survey factors and familiarity with the U.S. Food and Drug Administration’s term “medically important antimicrobial drugs”.

Variables	Coefficient	SE	Odds Ratio	95% CI		*p*-Value ^2^
				Lower	Upper	
Region ^1^						
GSCA	Referent					
NCA + NSJV	−0.49	0.51	0.61	0.22	1.63	0.32
Herd size, milking cows						
<1304	Referent					
≥1304	−1.10	0.63	0.33	0.09	1.17	0.09
Breed						
Holstein	Referent					
Jersey	−0.82	1.41	0.44	0.02	7.01	0.56
Crossbreed	1.75	1.25	5.74	0.49	66.25	0.16
Mix/Other	−0.37	0.59	0.69	0.21	2.22	0.54
Which AMD treatment information do you track or record?						
No milk or meat withdrawal interval	Referent					
Included milk and meat withdrawal interval	2.36	0.69	10.57	2.73	40.94	<0.01
Included milk or meat withdrawal interval	−0.34	0.99	0.71	0.10	5.01	0.74
Do you have a veterinarian–client–patient relationship (VCPR)?						
No	Referent					
Yes	2.73	1.24	15.29	1.34	173.53	0.03

^1^ Northern California (NCA), Northern San Joaquin Valley (NSJV), and Greater Southern California (GSCA). ^2^ Factors are statistically significant at *p* ≤ 0.05.

**Table 2 antibiotics-11-00165-t002:** Estimated coefficients and odds ratios from a multilevel mixed-effects logistic regression model for the association between survey factors and use or increased use of preventive alternatives to AMD.

Variables	Coefficient	SE	Odds Ratio	95% CI		*p*-Value ^2^
				Lower	Upper	
Region ^1^						
GSCA	Referent					
NCA + NSJV	−0.79	0.64	0.45	0.13	1.61	0.22
Herd size						
<1304	Referent					
≥1304	−0.75	0.77	0.47	0.10	2.13	0.32
Breed ^3^						
Holstein	Referent					
Others (Jersey, crossbreed, and mix)	0.68	0.68	1.97	0.51	7.61	0.32
Participate in any dairy quality assurance programs?						
No	Referent					
Yes	1.26	0.65	3.55	0.98	12.80	0.05
Changes made by farm regarding MIADs previously available OTC since 2018						
Increased/no changes	Referent					
Decreased	1.79	0.74	5.99	1.38	25.84	0.01

^1^ Northern California (NCA), Northern San Joaquin Valley (NSJV), and Greater Southern California (GSCA). ^2^ Factors are statistically significant at *p* ≤ 0.05. ^3^ Breed was categorized into two levels, namely Holstein and others (Jersey, crossbreed, and mixed herds), due to small sample size.

**Table 3 antibiotics-11-00165-t003:** Estimated coefficients and odds ratios from a multilevel mixed-effects logistic regression model for the association between antimicrobial drugs (AMD) survey factors and better farm animal health.

Variables	Coefficient	SE	Odds Ratio	95% CI		*p*-Value ^2^
				Lower	Upper	
Region ^1^						
GSCA	Referent					
NCA + NSJV	−0.30	0.51	0.76	0.28	2.10	0.59
Herd size						
<1304	Referent					
≥1304	−0.33	0.53	0.72	0.25	2.10	0.54
Breed ^3^						
Holstein	Referent					
Other (Jersey, crossbreed, and mix)	−0.02	0.55	0.97	0.32	2.90	0.96
Who decides AMD to treat sick cows?						
Dairy personnel only	Referent					
Veterinarian involved	−1.20	0.51	0.31	0.12	0.84	0.02
Have you used on-farm diagnostic techniques to guide AMD treatment?						
No	Referent					
Yes	1.51	0.52	4.53	1.61	12.71	<0.05
Have you made changes in management to prevent disease outbreak/spread?						
No	Referent					
Yes	1.10	0.52	2.91	1.10	8.10	0.04
Farm’s AMD costs						
Increased/no change	Referent					
Decreased	1.30	0.54	3.68	1.30	10.78	0.01

^1^ Northern California (NCA), Northern San Joaquin Valley (NSJV), and Greater Southern California (GSCA). ^2^ Factors are statistically significant at *p* ≤ 0.05. ^3^ Breed was categorized into two levels, namely Holstein and others (Jersey, crossbreed, and mixed herds), due to small sample size.

## Data Availability

This study was sponsored by the California Department of Food and Agriculture, and is subject to California Food and Agriculture Code (FAC) Sections 14400 to 14408. FAC Section 14407 requires that the data collected be kept confidential to prevent individual identification of a farm or business; as such, raw data from this study are not able to be shared.

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
