# Peer review of "Factors Associated with Antimicrobial Stewardship Practices on California Dairies: One Year Post Senate Bill 27"

_antibiotics, 2022, doi:10.3390/antibiotics11020165_

Round 1

Reviewer 1 Report

The manuscrip describes factors associated with the appropriate use of antibiotics on dairies, based on producer responses in a survey conducted one year post Senate Bill 27. I believe that the survey had very low participation and it is possible that only the producers most interested in the subject answered, so these two biases should be developed in depth in the itemstudy limitations.

The conclusions should be written in a better way in orden to clearly identify which is the best predictor for the appropriate use of antimicrobial and the point out the biases of the study in each conslusion.

In supplementary material, the survey questions and the ethics certificate should be attached.

Reviewer 2 Report

Thank you for sharing this interesting study on antimicrobial stewardship in California diaries. Although well written, it would be pleasing to the reader if the results and discussion sections are separate. The general structure of the manuscript including order of the sections may also be improved.

  • ‘2019 survey’ may not be necessary in the title.
  • Some of the information towards the end of the introduction (lines 85 to 92) may better suit the methodology section.
  • The methods section should be presented before the results.
  • How were the diaries that were sent questionnaires to determined?
  • What was the response rate?
  • A section on ethical considerations is needed.
  • Was reference made to any existing tools or policies while developing the questionnaire?
  • Was the tool piloted? And if so, where?
  • How was the sample size used determined?
  • It is advisable for the results and discussion sections to be written separately hence distinct.
  • Statistically significant results may be indicated in the various tables.
  • There is over reference to Ekong et al (who I believe is one of the authors). More literature may therefore be used in the manuscript.

Reviewer 3 Report

The manuscript addresses an important subject area relating to antimicrobial stewardship and medically important antimicrobial use in Californian dairy farms. Unfortunately the manuscript needs significant revisions and re-writing in order to consider it for publication. The results and discussion section is too long and includes too much detail that it is impossible to follow what the important messages are in the study. The authors need to review all of the results and either present these in more than one manuscript or present the main findings in one manuscript. The grammar is sometimes inconsistent and the use of abbreviations is confusing. Abbreviations are not always defined on first use which adds to the confusion. 

It is not clear what the definition of Medically Important Antimicrobial Drugs is that the authors are using. This seems to be the definition by the US Food and Drug Administration however, without defining this it is very confusing. The manuscript reads as if the reader has a good level of knowledge on US antimicrobial use practices and laws and needs to set the scene more clearly for a non-US audience. 

The formatting of the figures is not consistent with each other and Figure 1 is too busy and confusing. Overall, there are a lot of tables and the authors need to reduce the number of tables and make use of Supplementary Information for some of these.

The paper should be completely re-drafted with a different focus and reduced in length before being suitable for publication.

Round 2

Reviewer 2 Report

The authors have satisfactorily addressed the comments provided earlier.

Reviewer 3 Report

Many thanks to the authors for their extensive and thorough revision of the manuscript based on my previous comments. The manuscript now presents the study well in a clear and concise format. It will be of great interest to the readers of Antibiotics journal.